# A COVID-19 Silver Lining—Decline in Antibiotic Resistance in Ischemic Leg Ulcers during the Pandemic: A 6-Year Retrospective Study from a Regional Tertiary Hospital (2017–2022)

**DOI:** 10.3390/antibiotics13010035

**Published:** 2023-12-29

**Authors:** Amaraporn Rerkasem, Pak Thaichana, Nuttida Bunsermvicha, Rawee Nopparatkailas, Supapong Arwon, Saranat Orrapin, Termpong Reanpang, Poon Apichartpiyakul, Saritphat Orrapin, Boonying Siribumrungwong, Nongkran Lumjuan, Kittipan Rerkasem, José G. B. Derraik

**Affiliations:** 1Environmental-Occupational Health Sciences and Non-Communicable Diseases Research Center, Research Institute for Health Sciences, Chiang Mai University, Chiang Mai 50200, Thailand; amaraporn.rer@cmu.ac.th (A.R.); pak.th@cmu.ac.th (P.T.); 2Research Center for Infectious Disease and Substance Use, Research Institute for Health Sciences, Chiang Mai University, Chiang Mai 50200, Thailand; 3Clinical Surgical Research Center, Department of Surgery, Faculty of Medicine, Chiang Mai University, Chiang Mai 50200, Thailand; nuttida.b@cmu.ac.th (N.B.); supapong.arworn@cmu.ac.th (S.A.); saranat.orrapin@cmu.ac.th (S.O.); termpong.reanpang@cmu.ac.th (T.R.); poon.ap@cmu.ac.th (P.A.); 4Department of Family Medicine, Faculty of Medicine, Chiang Mai University, Chiang Mai 50200, Thailand; rawee.nop@cmu.ac.th; 5Department of Surgery, Faculty of Medicine, Thammasat University, Rangsit Campus, Pathum Thani 12120, Thailand; orrapins@staff.tu.ac.th (S.O.); boonying@tu.ac.th (B.S.); 6Research Center for Molecular and Cell Biology, Research Institute for Health Sciences, Chiang Mai University, Chiang Mai 50200, Thailand; nongkran.l@cmu.ac.th; 7Department of Women’s and Children’s Health, Uppsala University, 75185 Uppsala, Sweden; 8Department of Pediatrics: Child and Youth Health, Faculty of Medical and Health Sciences, University of Auckland, Auckland 1142, New Zealand

**Keywords:** bacteria, chronic limb-threatening ischemia, community health, ischemic leg ulcer, empirical antibiotic use, gram-negative, gram-positive, Gram staining, infection control, multidrug resistance, peripheral arterial disease, polymicrobial infection, public health, susceptibility

## Abstract

Antibiotic resistance (AR) associated with chronic limb-threatening ischemia (CLTI) poses additional challenges for the management of ischemic leg ulcers, increasing the likelihood of severe outcomes. This study assessed AR prevalence in bacteria isolated from CLTI-associated leg ulcers before (1 January 2017–10 March 2020; *n* = 69) and during (11 March 2020–31 December 2022; *n* = 59) the COVID-19 pandemic from patients admitted with positive wound cultures to a regional hospital in Chiang Mai (Thailand). There was a marked reduction in AR rates from 78% pre-pandemic to 42% during the pandemic (*p* < 0.0001), with rates of polymicrobial infections 22 percentage points lower (from 61% to 39%, respectively; *p* = 0.014). There were reduced AR rates to amoxicillin/clavulanate (from 42% to 4%; *p* < 0.0001) and ampicillin (from 16% to 2%; *p* = 0.017), as well as multidrug resistance (19% to 8%; *p* = 0.026). Factors associated with increased AR odds were polymicrobial infections (adjusted odds ratio (aOR) 5.6 (95% CI 2.1, 15.0); *p* = 0.001), gram-negative bacteria (aOR 7.0 (95% CI 2.4, 20.5); *p* < 0.001), and prior use of antibiotics (aOR 11.9 (95% CI 1.1, 128.2); *p* = 0.041). Improvements in infection control measures and hygiene practices in the community during the pandemic were likely key factors contributing to lower AR rates. Thus, strategic public health interventions, including community education on hygiene and the informed use of antibiotics, may be crucial in mitigating the challenges posed by AR in CLTI. Further, advocating for more judicious use of empirical antibiotics in clinical settings can balance effective treatment against AR development, thereby improving patient outcomes.

## 1. Introduction

Chronic limb-threatening ischemia (CLTI) is the advanced stage of atherosclerosis-induced chronic occlusive peripheral arterial disease (PAD) [1]. PAD results from severe blockage of lower-limb arteries, resulting in reduced blood perfusion; this may lead to tissue damage, unhealed ulcers, necrosis, and an increased susceptibility to infections [1,2,3]. The co-occurrence of infection and ischemia poses a substantial risk to the patient, potentially leading to major limb amputations or even fatality [4]. Severe limb infections extend to tendons, ligaments, bones, and even the proximal limb, requiring aggressive debridement and potentially leading to limb loss. As a result, CLTI has a 1-year mortality rate of 25%, and approximately 30% of CLTI patients require lower limb amputation [5].

Concurrent infections present at ulcer sites and the presence of necrotic tissue markedly affect treatment outcomes, particularly following revascularization surgery [6,7]. When superimposed infections occur in ischemic leg ulcers, treatment typically consists of multiple courses of high-potency, empirical, broad-spectrum antibiotics over an extended period before referral to a tertiary care center. However, the compromised blood flow in ischemic tissue hinders adequate antibiotic delivery, facilitating polymicrobial infections and the proliferation of highly virulent organisms [8]. Consequently, superimposed infections in ischemic leg ulcers increase the risk of multidrug-resistant (MDR) pathogens occurring [9,10], potentially leading to life-threatening untreatable infections [8]. Additionally, MDR pathogens can evolve into extensively drug-resistant strains, requiring aggressive treatment regimens involving combinations of different drugs [11,12,13].

A systematic review and meta-analysis included 148 studies and approximately 363,000 patients covering 2019–2021 [14]. It reported a 5% prevalence of bacterial co-infections and an 18% prevalence of secondary bacterial infections among hospitalized COVID-19 patients [14]. However, only 42 of the included studies reported comprehensive data on antimicrobial susceptibility, with 61% of recorded bacterial infections involving resistant pathogens [14]. Another meta-analysis of 23 studies yielded inconclusive results [15], but there have been reports of increased antibiotic resistance during the COVID-19 pandemic [16,17]. In contrast, several studies have reported reductions in antibiotic-resistance prevalence during the pandemic [18,19,20]. The uncertainties regarding COVID-19 diagnosis and concerns over bacterial co-infections or secondary infections likely underpinned substantial but inappropriate prescription of antibiotics [15]. While the latter might have led to increased antibiotic resistance, the evidence remains conflicting and, thus, inconclusive. 

For the physician, ischemic leg ulcers are a concern as they frequently lead to severe pain, systemic inflammation, and a high risk of treatment failure, potentially leading to hospitalization and the need for both antibiotic therapy and surgical interventions [3]. In 2020–2022, during the COVID-19 pandemic, inevitable delays occurred in accessing minor clinical procedures, such as special dressing or debridement, due to the strict screening protocols that restricted hospital entry and exit and limited non-urgent admissions [21]. The consequences of delayed treatment were exacerbated by the demands on medical resources for COVID-19 patients, which put substantial pressure on healthcare systems [22,23]. In this context, physicians frequently resorted to empiric broad-spectrum antibiotics to mitigate the risks of deep infection and potential limb loss. However, this practice led to adverse public health consequences, such as variations in antibiotic-resistant genes, as well as changes in the spectrum of pathogens and the severity of associated foot infections. Conversely, it was suggested that the COVID-19 pandemic would not substantially increase the prevalence of antimicrobial resistance in most countries, owing to enhanced infection prevention and control efforts in community and healthcare settings [24].

Although several studies have explored changes in antibiotic resistance associated with diabetes-related foot ulcers during the COVID-19 pandemic [16,21], to our knowledge, no studies have specifically reported the prevalence and spectrum of bacteria isolated from ischemic leg ulcers. Therefore, we examined the infection rates of antibiotic-resistant bacteria in ischemic leg ulcers among PAD patients admitted to a tertiary medical center before and during the COVID-19 pandemic. In addition, we aimed to characterize the recorded bacterial taxa, including the antibiotics they were resistant to. Lastly, we explored the associated risk factors and characteristics of the infected wounds.

## 2. Results

### 2.1. Study Population

During the six years covered by this study, 197 patients were admitted with CLTI of Rutherford class ≥5, including 106 patients pre-COVID-19 and 91 during the pandemic, but 37 and 32 patients, respectively, were subsequently excluded (Figure 1). Thus, 69 and 59 patients, respectively, met the main inclusion criteria of confirmed chronic ischemic leg ulcers with signs of infection and positive wound cultures (Figure 1).

The demographic characteristics of the study population are shown in Table 1. Most patients were male (62%) and elderly, with a median age of 69 years (range 35–99 years) (Table 1). Pre-pandemic and pandemic patients had similar demographic and clinical characteristics, except for higher rates of revascularization procedures during the COVID-19 pandemic (51 vs. 32%; *p* = 0.029). There was also indication that the median duration of ulceration before hospitalization was longer in the pre-pandemic compared to the pandemic period (30 vs. 20 days, respectively; *p* = 0.052) (Table 1). Importantly, rates of systemic inflammatory response syndrome (SIRS) and severity of wound infections were similar in the two periods (Table 1).

### 2.2. Infection Types and Prescribed Antibiotics

Microbiological analyses showed a markedly higher rate of polymicrobial infections pre-pandemic compared to the pandemic period (61% vs. 39%, respectively, or +22% (95% CI 5, 39%); *p* = 0.014) (Table 2). All but one patient was prescribed an antibiotic at admission (Table 2). Over two-thirds of patients (68%) across the six years had exclusive gram-negative infections, with similar rates in the two periods (69% vs. 67%, respectively; Table 2). However, rates of mixed infections were more than two-fold higher pre-pandemic (23% vs. 10% or +13.0% (95% CI 0.4, 26%); *p* = 0.043).

Overall, 95% of patients received empirical antibiotic treatment before wound culture, with little change between periods (Table 2). Among the empirical treatment regimens, clindamycin and ciprofloxacin were the most prescribed antibiotics (75% and 69%, respectively), without evidence of rate differences between periods (Table 2). Similarly, combined treatment rates with ciprofloxacin and clindamycin did not change (62% pre-pandemic and 64% during the pandemic).

### 2.3. Differences in AR Rates

The six-year rate of patients with at least one bacterial taxon with AR cultured from a leg ulcer was 61.7% (79/128). However, there were marked differences between study periods (Figure 2). Pre-pandemic, the three-year AR rate was 78% (54/69) compared to 42% (25/59) during the COVID-19 pandemic (+36% (95% CI 20, 52%); *p* < 0.0001) (Figure 2). Notably, while there was relatively limited variation in yearly AR rates pre-pandemic, rates progressively increased throughout the COVID-19 pandemic (29% in 2020, 43% in 2021, and 50% in 2022; Figure 2). From the 128 patients, 225 bacterial samples were isolated, of which 79.1% were gram-negative and 20.9% were gram-positive.

When data on the 225 bacterial samples tested were compared between the two periods, pre-pandemic AR rates for amoxicillin/clavulanate were 10-fold higher (42% vs. 4%; +38% (95% CI 27, 51%); *p* < 0.0001) and for ampicillin 8-fold higher (16% vs. 2%; +14% (95% CI 5, 24%); *p* = 0.017) (Table 3). Despite the proportionally lower numbers of antibiotic-resistant isolates during the pandemic for seven of the 12 antibiotics tested, no other statistically significant differences were observed between the two periods (Table 3).

There were 33 bacterial isolates (14.7%) considered of greater clinical and public health relevance (Table 4). Consistent with the above-described findings, the pre-pandemic 3-year rate of MDR bacteria was markedly higher compared to the pandemic rate (18.8% vs. 8.0%; +10.8% (95% CI 2.1, 19.5%); *p* = 0.026) (Table 4). Nonetheless, most of these bacteria were relatively rare in our study population, ranging from 0.4% to 1.3%, except for the higher prevalence of ESBL *Escherichia coli* (4.9%; 11/225), ESBL *Proteus* spp. (3.6%; 8/225), and ESBL *Klebsiella* spp. (3.1%; 7/225) (Table 4).

### 2.4. Gram-Positive Bacteria

The analysis of the 47 samples of gram-positive bacteria showed that overall AR rates in the pre-pandemic and pandemic periods were 29% and 22%, respectively (Table 5). Although the proportion of antibiotic-resistant isolates seemed to be lower for four of the ten drugs tested during the pandemic, no statistically significant differences were detected between the two periods (Table 5).

The overall AR rate among gram-positive bacteria was 26% (12/47), with at least one isolated taxon displaying AR against one out of 10 antibiotics (Table 5 and Appendix A). *Staphylococcus aureus* was the predominant gram-positive bacterium isolated (40%; 19/47), with 16% (3/19) of cultures found to be antibiotic-resistant (Appendix A and Table 5). *Enterococcus* spp. was the second most common gram-positive taxon (*n* = 17; 36%) with a 35% AR rate (6/17; Appendix A and Table 5). These two taxa comprised 77% (36/47) of gram-positive isolates (Appendix A and Table 5). The antibiotics with the greatest AR rates among gram-positive bacteria were clindamycin and erythromycin (both 17% (5/30); Appendix A and Table 5). Among the two *Staphylococcus epidermidis* isolates, AR was observed against six of the seven antibiotics tested (Appendix A and Table 5).

### 2.5. Gram-Negative Bacteria

In Table 6, the 178 gram-negative bacterial samples tested were compared between the two periods. The overall pre-pandemic AR rate was higher than the pandemic rate (61% vs. 39%, respectively; +22% (95% CI 7, 37%); *p* = 0.007). This observation aligns with the markedly higher pre-pandemic AR rates for specific antibiotics: ampicillin was 19-fold higher (19% vs. 0%; +19% (95% CI 9, 29%); *p* = 0.006) and amoxicillin/clavulanate was 10-fold higher (42% vs. 4%; +38% (95% CI 27, 51%); *p* < 0.0001) (Table 6). Among individual gram-negative bacteria, there was a detectable AR rate reduction for *Morganella morganii* despite very few pandemic isolates (*n* = 3), as all ten pre-pandemic isolates tested were antibiotic-resistant (33% vs. 100%; +67% (95% CI 13, 100%); *p* = 0.014) (Table 6).

Over half of the individual gram-negative bacterial cultures (53%; 94/178) were resistant to at least one of the eight antibiotics tested (Table 6 and Appendix A). This AR rate was markedly higher compared to the observed 26% rate for gram-positive bacteria (+27% (95% CI 13, 42%); *p* < 0.001). Further, 12 out of 13 gram-negative bacterial taxa in the study were resistant to at least two antibiotics (Appendix A and Table 6). *Escherichia coli* and *Pseudomonas aeruginosa* were the most common gram-negative bacteria isolated, each making up 15% of the overall records for this group (27/178 and 26/178, respectively; Appendix A and Table 6). Notably, the AR rate against at least one antibiotic for both *Escherichia coli* and *Morganella morganii* was 85% (23/27 and 11/13, respectively), with 77% of *Morganella morganii* found to be resistant against amoxicillin/clavulanate (10/13) (Appendix A and Table 6). The highest AR rates were recorded against fluoroquinolones (30%; 53/178) and amoxicillin/clavulanate (29%; 39/134) (Appendix A and Table 6). Importantly, among *Escherichia coli*, AR was recorded against all eight antibiotics tested and against seven antibiotics for *Citrobacter freundii* and *Klebsiella pneumoniae* (Appendix A and Table 6).

### 2.6. AR Risk Factors

In univariable analyses, the odds ratio of AR during the COVID-19 pandemic was nearly five-fold lower compared to the pre-pandemic period (OR 0.21; *p* < 0.001) (Table 7). Other risk factors associated with increased odds of AR were having a polymicrobial infection (OR 3.37 vs. monomicrobial; *p* = 0.002) or a wound infection with gram-negative bacteria alone (OR 2.57; *p* = 0.015) and hospitalization within 6 months before admission (OR 2.10; *p* = 0.038) (Table 7).

The multivariable logistic regression yielded adjusted odds ratios (aOR) identical to the univariable analysis, also showing odds of AR approximately five-fold lower during the pandemic period (0.21 (95% CI 0.08, 0.51); *p* = 0.001) (Table 7). Risk factors that remained associated with increased odds of AR were a polymicrobial infection (aOR 5.58; *p* = 0.001) or a gram-negative only infection (aOR 6.98; *p* < 0.001) (Table 7). In addition, there were markedly higher odds of AR in association with previous empirical antibiotic treatment (aOR 11.9; *p* = 0.041) and increased respiratory rate at admission (aOR 6.17; *p* = 0.047) (Table 7).

## 3. Discussion

The COVID-19 pandemic has profoundly affected healthcare services worldwide, particularly regarding resource distribution and, thus, patient care. However, this study shows that the pandemic was associated with markedly lower AR rates among patients admitted to our tertiary hospital in Chiang Mai with infected ischemic leg ulcers due to CLTI. These were accompanied by considerably lower AR rates for certain antibiotics (e.g., amoxicillin/clavulanate and ampicillin) and lower rates of polymicrobial and mixed infections, while the rate of MDR infections more than halved. The latter is particularly notable as MDR bacteria are associated with more severe infections and worse patient outcomes [6].

Our observations are corroborated by previous studies consistently showing decreases in AR rates among inpatients during the pandemic, such as in Italy [18,20], Turkey [19], and Singapore [25]. These were largely attributed to improved hospital infection prevention and control measures [18,19,20]. Practices that reportedly mitigate the risk of nosocomial infections with AR pathogens include screening potential COVID cases, segregating patients with respiratory symptoms, enforcing physical distancing between beds, mandating masks, and intensifying disinfection protocols [25]. Further evidence that lower AR rates in Chiang Mai were likely due to COVID-19 measures was provided by the steady increase in yearly AR rates throughout the 3-year study period following the pandemic declaration in March 2020.

Importantly, apart from the pandemic-associated reduction in rates of nosocomial infections (for pathogens other than SARS-CoV-2), our study also indicated reductions in home- and community-acquired AR infection rates. Our samples were collected directly from wound sites at the time of patient admission, which indicates frequent bacterial colonization of wounds acquired in the community. Thus, it is likely that many CLTI infections with secondary complications originate from community settings. Several factors may contribute to community-acquired AR infections, such as inadequate wound care, inadequate use and/or overuse of antibiotics [26], exposure to infected sources, and poor hygiene practices [27]. In addition, AR bacteria can also increase in community settings via contaminated food and water [28].

Additional strong evidence of marked reductions in community transmission of common pathogens during the pandemic was reported in New Zealand [29]. During the winter of 2020, stringent government-imposed lockdowns led to the near complete elimination of influenza virus transmission, when surveillance data showed a 99.9% reduction in influenza cases compared to previous years [29]. Our study’s apparent reduction in community-acquired infections with antibiotic-resistant bacteria may be partially attributed to government-imposed measures and heightened public awareness following the COVID-19 pandemic declaration in March 2020 [30]. In Thailand, the government imposed a range of measures, including the inception of the Centre for COVID-19 Situation Administration (CCSA), the imposition of lockdowns, border controls, quarantine protocols, and public hygiene campaigns [30]. The success of the collaborative effort between government initiatives and public compliance halted community transmission of SARS-CoV-2 during the first wave, achieving 220 consecutive days with no confirmed cases of community transmission [31]. While the CCSA was officially closed in October 2020 and lockdowns were lifted, several key preventative measures remained in place to some extent throughout the three-year duration of the pandemic. These included physical distancing, mask-wearing, regular handwashing, frequent antigen self-testing, and continuous risk communication to the public [31].

Clinical practice guidelines for ischemic leg infections are mostly based on studies of diabetic foot infections [32]. Here, we examined bacterial pathogens associated with CLTI, which are less explored. Our results suggest different infection patterns in CLTI-associated leg ulcers compared to diabetic foot infections. Regarding Gram stain characteristics, diabetic foot infections appear to be colonized mostly by gram-positive bacteria (≈75%) [10,32] compared to 21% among our patients and ≈42% in a Romanian study [33]. Notably, early-stage wounds are mainly infected by gram-positive bacteria, such as *Staphylococcus* spp. and *Streptococcus* spp., which come from normal skin flora [34]. In contrast, chronic wounds, if deprived of oxygen due to arterial insufficiency, can lead to prolonged infections, often colonized by gram-negative bacterial genera such as *Pseudomonas*, *Acinetobacter*, and *Stenotrophomonas* [35]. In our study, *Staphylococcus aureus* and *Enterococcus* spp. were the main gram-positive bacteria recorded. Still, infections were primarily associated with gram-negative bacteria, particularly *Escherichia coli* and *Pseudomonas aeruginosa*.

The choice of antibiotic treatment in CLTI often relies on expert opinion rather than evidence-based recommendations [6,33]. While there seems to be no consensus on the optimal antibiotic regimen for CLTI, empirical antibiotic therapy is the most common approach for such infections, targeting the most likely pathogens before the exact bacterium species is identified. However, these infections can lead to complications such as failed revascularization, resulting in the incomplete healing of ischemic lesions and subsequently increasing the risk of major limb amputations [36,37]. A previous study showed that infected CLTI patients who received empirical antibiotics before revascularization had amputation and death rates similar to those without infection [36]. Therefore, it is concerning that we observed relatively high AR rates for commonly used empirical antibiotics for CLTI, especially ciprofloxacin, clindamycin, and amoxicillin/clavulanate. Antibiotic resistance to fluoroquinolone and amoxicillin/clavulanate was approximately 30% among gram-negative bacteria. While robust conclusions cannot be drawn from routinely collected retrospective data, 95% of our patients hospitalized with CLTI reported a history of previous antibiotic treatment, mostly prescribed by general practitioners. Therefore, our findings suggest that excessive use of empirical antibiotics was likely an important contributor to antibiotic resistance in CLTI, especially when empirical therapy involves ineffective antibiotics against the causative organism(s) [10]. As a result, future studies should compare previous empirical antibiotic treatments with the laboratory results of antibiotic resistance in patients with CLTI.

A limitation of our study was its retrospective design involving a single center with a relatively limited number of cases. While our hospital served as a tertiary center during the pandemic, we faced a manageable number of COVID-19 inpatients compared to secondary or provincial hospitals. Therefore, caution is required when generalizing our results. In addition, our cross-sectional study examined data collected only at the patient’s hospital admission, which might have biased our observations regarding antibiotic-resistance rates in the absence of longitudinal data. Further, we did not evaluate how different wound dressing and care approaches might have affected secondary infections. Lastly, several bacterial taxa (e.g., *Citrobacter freundii* and *Klebsiella pneumoniae*) exhibit natural resistance to specific antibiotics [38,39], which would likely affect susceptibility assessments, as reported, for example, for *Proteus vulgaris* and *Enterobacter cloacae* to aminoglycosides and *Acinetobacter baumannii* to fluoroquinolones [38,40,41]. However, distinguishing between natural and acquired resistance is challenging. While underlying natural resistance is most unlikely to have yielded Type I errors (i.e., concluding that AR rates changed when they did not), such bacterial taxa might have masked certain changes in AR rates, leading to Type II errors. Despite these considerations, our research is among the initial efforts to compare AR in ischemic limbs pre- and post-COVID-19. Additionally, comprehensive descriptions of antibiotic resistance and the related bacterial taxa in CLTI-associated leg ulcer infections within Thailand had not been previously explored.

## 4. Conclusions

We observed markedly lower rates of antibiotic resistance among patients admitted to our regional hospital with infected ischemic leg ulcers due to CLTI. We speculate that improvements in infection control measures and hygiene practices in the community during the pandemic were likely key factors contributing to lower AR rates. Thus, strategic public health interventions, including community education on hygiene and the informed use of antibiotics, may be crucial in mitigating the challenges posed by antibiotic resistance in CLTI. Further, advocating for more judicious use of empirical antibiotics in clinical settings can balance effective treatment against AR development, thereby improving patient outcomes.

## 5. Material and Methods

### 5.1. Ethics Approval

This study was approved by the Research Ethics Committee of the Faculty of Medicine, Chiang Mai University (SUR-2565-09322). Informed consent from individual patients was not necessary, as this was a retrospective audit of data from routine clinical care based on de-identified patient data.

### 5.2. Study Design

This study was conducted at the Maharaj Nakorn Chiang Mai Hospital, a teaching hospital affiliated with the Faculty of Medicine at Chiang Mai University (Chiang Mai, Thailand). It was the first university hospital established outside Bangkok, and it is an important regional medical center providing tertiary care in Northern Thailand. This hospital is well-equipped to manage complex medical cases, including CLTI and AR. Patient data for this study were sourced from the Department of Vascular Surgery, spanning six years from 1 January 2017 to 31 December 2022.

We extracted data from electronic medical records, including admission and discharge summaries, laboratory data, and microbiological diagnostic information for infected ulcers. Eligible participants were all patients with PAD assessed for CLTI and diagnosed with chronic ischemic leg ulcers using Rutherford categories before admission [3]. PAD was diagnosed based on abnormal pedal pulse palpation due to chronic atherosclerosis and confirmed by poor arterial perfusion using at least one of the following methods [1,42,43]: ▪ankle-brachial index (ABI)▪toe pressure measurements▪transcutaneous oxygen measurement (TCOM)▪color Doppler ultrasound▪computed tomographic angiography (CTA)

The 6-stage Rutherford classification was used to categorize PAD severity [1]: ▪0—asymptomatic▪1—mild claudication▪2—moderate claudication▪3—severe claudication▪4—ischemic rest pain▪5—minor tissue loss▪6—major tissue loss

CLTI, indicative of advanced PAD stage, was defined as persistent foot pain or tissue necrosis for at least two weeks. In this study, ischemic leg ulcers were specifically defined as CLTI cases classified under the Rutherford Classification stage ≥ 5. Hemodynamic criteria for CLTI included [3,44]:▪resting ankle systolic pressure < 50–70 mmHg▪toe pressure < 40 mmHg in non-diabetic or <50 mmHg in diabetic patients▪TCOM < 30 mmHg

Using the guidelines from the Infectious Diseases Society of America/International Working Group on Diabetic Foot (IDSA/IWGDF) [32], we classified the severity of infected ischemic wounds based on the presence of at least two of the following:▪local swelling or induration▪erythema (redness) >0.5 cm around the wound▪local tenderness or pain▪local hyperthermia▪purulent discharge not attributable to other conditions (e.g., trauma, gout, or venous stasis

The depth of infection into bone structures was evaluated using the probe-to-bone test, with positive results leading to further investigation by X-ray for osteomyelitis diagnosis. The Society of Vascular Surgery’s Wound, Ischemia, and Foot Infection (SVS-WIfI) classification system [2] was used to evaluate the overall severity of ulcers and infections. Information regarding patient demographics, medical history, clinical evaluation, and diagnosis were extracted at the first admission date. Revascularization refers to open surgical bypass, endarterectomy, and endovascular procedure. Minor limb amputations were defined as the transverse removal of any part of the lower limb at the level of the ankle joint or below; major amputations were classified as any amputation above the ankle joint up to the thigh [32,45].

For patients diagnosed with CLTI multiple times throughout the study period, only the data from their first admission were included in this study. Other exclusion criteria were the absence of systemic or local symptoms, lack of signs of infection alongside positive wound cultures, and instances where specimens were contaminated during handling and/or processing.

### 5.3. Specimens and Microbiology

Bacterial pathogens were primarily identified through microbiological tests on pus samples or wound biopsies collected within the first few days of admission and before the administration of antibiotics at our hospital. It should be noted, however, that most patients had likely already received empirical antibiotics from primary healthcare providers before admission to our facility. We included only the first isolates per patient to prevent the overestimation of microbiological isolations resulting from multiple samples within the same patients. Inconsistencies were resolved using wound biopsy results. Bacterial identification was performed using MALDI-TOF [46], and antibiotic susceptibility was tested using the Vitek2 system (bioMerieux, Marcy L’Etoile, France). The results were interpreted according to the Clinical and Laboratory Standards Institute (CLSI) guidelines [47].

The levels of antibiotic resistance of each identified pathogen were categorized into susceptible, intermediate, and resistant phenotypes, with intermediate results treated as susceptible. We assessed AR risks across various antibiotic groups and identified numerous bacterial pathogens resistant to antibiotics commonly used in routine clinical practice. These pathogens pose a considerable threat to public health, leading to prolonged illness and, in some cases, death, as well as increased healthcare costs. To assess the risk of AR across antibiotic groups (penicillins, clindamycin, macrolides, sulfonamides, cephalosporins, quinolones, and aminoglycosides), we considered a loss of susceptibility affecting the bactericidal or bacteriostatic properties of an antibiotic agent, as indicated by laboratory results [8,12,16]. Subsequently, we analyzed the most frequently isolated agents over the study period, including Methicillin-resistant *Staphylococcus aureus* (MRSA), Methicillin-resistant *Staphylococcus epidermidis* (MRSE), extended-spectrum β-lactamase (ESBL)-producing bacteria (including those resistant to third-generation cephalosporin), Vancomycin-resistant *Enterococcus* (VRE), Carbapenem-resistant Enterobacterales (CRE, including *Escherichia coli*, *Klebsiella* spp., and *Proteus* spp.), Carbapenem-resistant *Pseudomonas* spp., and Carbapenem-resistant *Acinetobacter* spp. Multidrug resistance (MDR) was defined as acquired non-susceptibility to at least one agent in a minimum of three antibiotic categories, following the criteria established by the Clinical Laboratory Standards Institute [13].

### 5.4. Outcomes

The rates of antibiotic resistance in infected CLTI patients were compared between the COVID-19 pandemic and pre-pandemic years, determined based on the WHO’s declaration on 11 March 2020 [48]. The primary outcome was the rate of AR, defined as the number of patients with at least one antibiotic-resistant pathogen in their infected leg wound, stratified by the period before and during the COVID-19 pandemic. We selected the potential risk factors likely to influence AR outcomes independently, as previously reported [9,49]. These factors were assessed at hospital admission and coded as binary variables to simplify the comparison of descriptors and AR results: gender (male vs. female); age (≥60 vs. <60 years); diabetes mellitus (with vs. without); renal replacement therapy (undergoing vs. not undergoing); fever (≤38 vs. >38 °C) [32]; respiratory rate (≤20 vs. >20 breaths/min) [32], leukocytosis (≤12,000 vs. >12,000 cells/mm^3^) [32]; recurrent ulcer on the same limb (yes vs. no); duration of ulcer (<3 vs. ≥3 months); limb infection grade(<3 vs. >3) [32]; wound grade (<3 vs. 3) [32]; ischemic grade (<3 vs. 3) [32]; osteomyelitis (with vs. without) [32]; hospitalization within six months before admission (yes vs. no); referral case (yes vs. no); infection type (polymicrobial vs. monomicrobial); Gram stain (negative or positive vs mixed infection); any empirical antibiotic treatment in the last 30 days (yes vs. no).

### 5.5. Statistical Analysis

Differences in baseline clinical data between the two periods were evaluated using two-sample t-tests or the non-parametric Mann–Whitney U test for continuous data and chi-squared tests or Fisher’s exact tests for categorical data, as appropriate. The AR rates before and during the COVID-19 pandemic were compared using Fisher’s exact tests. The 95% confidence intervals (CI) for the differences between the two rates were derived using the normal approximation test for two proportions. Continuous data approximating a normal distribution were reported as the mean ± standard deviation (SD). Skewed data were reported as the median [quartile 1, quartile 3]. A multivariable logistic regression was used to explore the risk factors for infected ischemic leg ulcers in CLTI patients with antibiotic-resistant bacteria. A backward stepwise selection procedure was used to identify the predictors associated with AR likelihood. Variables with a *p*-value ≥ 0.1 were removed from the model at each step, and variables with a *p*-value < 0.05 were retained. Multicollinearity among risk factors was checked by the variance inflation factor. All tests were 2-sided with statistical significance set at *p* < 0.05. Statistical analyses were performed using STATA (16.1, Stata Corp LLC, College Station, TX, USA) and Minitab v21.4 (Pennsylvania State University, State College, PA, USA).

## Figures and Tables

**Figure 1 antibiotics-13-00035-f001:**
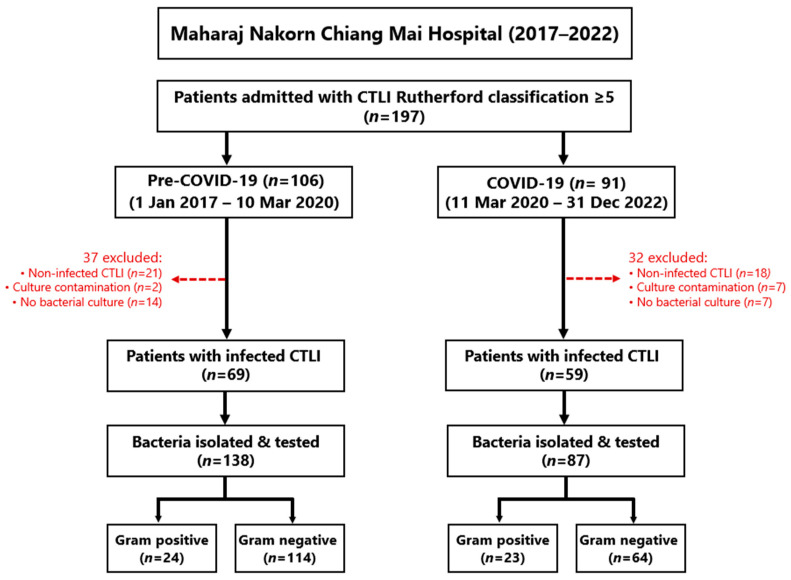
Flow diagram showing the recruitment of patients into this study, including the reasons for exclusion.

**Figure 2 antibiotics-13-00035-f002:**
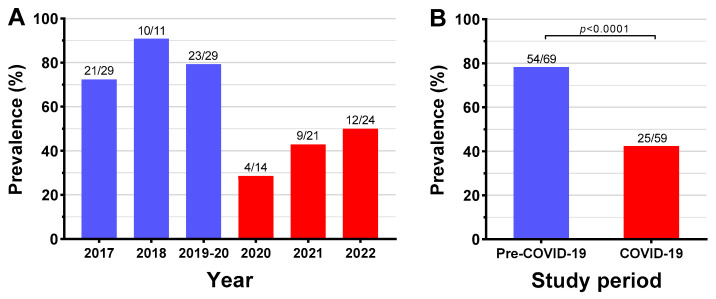
Yearly (**A**) and overall (**B**) prevalence of antibiotic resistance in 128 patients with CLTI-associated leg ulcers before (1 January 2017–10 March 2020) and during (11 March 2020–31 December 2022) the COVID-19 pandemic.

**Table 1 antibiotics-13-00035-t001:** Demographic and clinical characteristics of patients admitted to hospital with CLTI-associated ischemic leg ulcers before (1 January 2017–10 March 2020) and during (11 March 2020–31 December 2022) the COVID-19 pandemic.

Parameters		Overall	Pre-COVID-19	COVID-19	*p* ^A^
** *n* **		128	69	59	
**Age (years)**		69 [61, 77]	67 [60, 76]	70 [61, 78]	0.45
**Male**		79 (62%)	37 (54%)	42 (71%)	0.88
**Major atherosclerotic risk factors**					
Diabetes mellitus		83 (65%)	41 (59%)	42 (71%)	0.17
Hypertension		93 (73%)	49 (71%)	44 (75%)	0.65
Dyslipidemia		47 (37%)	22 (32%)	25 (42%)	0.22
Current smoking		6 (5%)	3 (4%)	3 (5%)	>0.99
Chronic kidney disease (stages III, IV, V) ^B^		68 (53%)	39 (57%)	29 (49%)	0.41
Receiving RRT		33 (26%)	20 (29%)	13 (22%)	0.37
**Manifestation of SIRS** ^C^					
Temperature >38 °C		25 (20%)	13 (19%)	12 (20%)	0.83
Tachycardia >90 bpm		46 (36%)	27 (39%)	19 (32%)	0.42
Tachypnea >20 bpm		14 (11%)	10 (14%)	4 (7%)	0.26
Abnormal leukocytes ^D^		52 (41%)	25 (36%)	27 (46%)	0.27
Positive SIRS score		48 (38%)	25 (36%)	23 (39%)	0.75
**CLTI leg ulcer parameters**					
History of previous ulcers (recurrent ulcer)		40 (31%)	22 (32%)	18 (31%)	0.87
Duration of active ischemic ulcer/gangrene (days)		30 [14, 68]	30 [14, 90]	20 [14, 60]	0.052
**Severity of ulcer and infection** ^E^					
Wound grade	1	9 (7%)	8 (11%)	1 (2%)	0.058
	2	77 (60%)	37 (54%)	40 (68%)	
	3	42 (33%)	24 (35%)	18 (30%)	
Ischemic grade	1	46 (36%)	28 (41%)	18 (30%)	0.10
	2	36 (28%)	22 (32%)	14 (24%)	
	3	46 (36%)	19 (27%)	27 (46%)	
Infection grade	1	15 (12%)	8 (12%)	7 (12%)	0.99
	2	71 (55%)	38 (55%)	33 (56%)	
	3	42 (33%)	23 (33%)	19 (32%)	
Clinical stage	2	3 (2%)	3 (4%)	–	0.41
	3	6 (4%)	3 (4%)	3 (5%)	
	4	119 (93%)	63 (92%)	56 (95%)	
Presence of osteomyelitis		41 (32%)	25 (36%)	16 (27%)	0.27
Presence of wound gangrene		121 (95%)	64 (93%)	57 (97%)	0.45
**Hospitalization and treatments**					
Any hospitalizations within 6 months		62 (48%)	38 (55%)	24 (41%)	0.10
Referral from health services for CLTI		40 (31%)	25 (36%)	15 (25%)	0.19
Surgical treatments at this admission					
Debridement		50 (39%)	23 (33%)	27 (46%)	0.15
Revascularization		52 (41%)	22 (32%)	30 (51%)	**0.029**
Underwent amputation		122 (95%)	66 (96%)	56 (95%)	>0.99
Amputation levels	Minor	102 (84%)	56 (85%)	46 (82%)	0.69
	Major	20 (16%)	10 (15%)	10 (18%)	

Continuous data are reported as median [quartile 1, quartile 3] and categorical data as *n* (%). CLTI, chronic limb-threatening ischemia; RRT, renal replacement therapy; SIRS, systemic inflammatory response syndrome. ^A^ Derived from chi-square tests, two-sample t-test, or non-parametric Mann–Whitney U comparing the two study periods, with the *p*-value for the statistically significant differences (at *p* < 0.05) shown in bold. ^B^ Staging of chronic kidney disease defined by an eGFR reduction <60 mL/min/1.73 m^2^. ^C^ Patients were diagnosed with SIRS when they met any two or more of the following criteria: body temperature, heart rate, respiratory rate, and white blood cell count. ^D^ Leukocyte count: leukopenia (<4000/mm^3^), leukocytosis (>12,000/mm^3^), or ≥10% immature neutrophils (band forms). ^E^ The SVS-WIfI classification grades CLTI based on perfusion (0–3), wound extent (0–3), and superadded infection (0–3), with 0 indicating absence and 3 indicating severe; the total score (summing these components) is associated with the risk of major amputations.

**Table 2 antibiotics-13-00035-t002:** Bacterial infection types and prescribed antibiotics in patients admitted to hospital with CLTI-associated ischemic leg ulcers before (1 January 2017–10 March 2020) and during (11 March 2020–31 December 2022) the COVID-19 pandemic.

Parameters		*N*	Pre-COVID-19	COVID-19	*p* ^A^
** *n* **		128	69	59	
**Infection type**					
Monomicrobial		63 (49%)	27 (39%)	36 (61%)	**0.014**
Polymicrobial		65 (51%)	42 (61%)	23 (39%)	
**Gram staining**					
Only gram-positive		19 (15%)	7 (10%)	12 (20%)	0.067
Only gram-negative		87 (68%)	46 (67%)	41 (70%)	
Mixed infection		22 (17%)	16 (23%)	6 (10%)	**0.043** ^D^
**Empirical antibiotics**					
Previous treatment ^B^		121 (95%)	66 (96%)	55 (93%)	0.70
Prescription by	General practitioner	103 (85%)	55 (83%)	48 (87%)	0.54
	Specialist	18 (15%)	11 (17%)	7 (13%)	
Treatment during this study ^C^	Any	127 (99%)	68 (99%)	59 (100%)	>0.99
	Clindamycin	96 (75%)	49 (71%)	47 (80%)	0.26
	Ciprofloxacin	83 (69%)	44 (64%)	39 (66%)	0.78
	Penicillin	10 (8%)	6 (9%)	4 (7%)	0.75
	Cephalosporin	15 (12%)	6 (9%)	6 (15%)	0.43

Data are *n* (%). ^A^ Derived from chi-square or normal approximation tests comparing the two study periods, with the *p*-value for the statistically significant difference (*p* < 0.05) shown in bold. ^B^ Antibiotic treatment information refers to the period before the first infection included in this study. ^C^ The antibiotics used were not mutually exclusive; the same patient might have received more than one. ^D^ Comparing mixed infections vs. others.

**Table 3 antibiotics-13-00035-t003:** Prevalence of antibiotic resistance in bacteria isolated from CLTI-associated leg ulcers before (1 January 2017–10 March 2020) and during (11 March 2020–31 December 2022) the COVID-19 pandemic.

Antibiotic	Overall	Pre-COVID-19	COVID-19	Difference	*p*
Oxacillin	3/22 (14%)	1/9 (11%)	2/13 (15%)	−4% (−33, 24%)	>0.99
Ampicillin	13/116 (11%)	12/73 (16%)	1/43 (2%)	14% (5, 24%)	**0.017**
Clindamycin	5/30 (17%)	3/14 (21%)	2/16 (13%)	8% (−19, 35%)	0.64
Erythromycin	5/30 (17%)	3/14 (21%)	2/16 (13%)	8% (−19, 35%)	0.64
Carbapenems	11/169 (7%)	6/107 (6%)	5/62 (8%)	−2% (−10, 6%)	0.53
Cephalosporins	32/178 (18%)	25/114 (22%)	7/64 (11%)	11% (0, 22%)	0.12
Piperacillin/tazobactam ^A^	8/169 (5%)	6/107 (6%)	2/62 (3%)	3% (−2, 9%)	0.49
Fluoroquinolones	54/224 (24%)	35/137 (26%)	19/87 (23%)	3% (−8, 15%)	0.62
Aminoglycoside	19/215 (9%)	9/130 (6%)	10/85 (13%)	−7% (−15, 1%)	0.28
Amoxicillin/clavulanate ^A^	39/134 (29%)	37/89 (42%)	2/45 (4%)	38% (27, 51%)	**<0.0001**
Trimethoprim/sulfamethoxazole ^A^	41/173 (24%)	25/110 (23%)	16/63 (25%)	−2% (−15, 11%)	>0.99
Penicillin	2/24 (8%)	1/14 (7%)	1/10 (10%)	−3% (−26, 20%)	>0.99

Data for individual periods or the overall study represent the proportions of isolated pathogens with resistance to a given antibiotic out of those tested. Antibiotic resistance rates between the two periods were compared using Fisher’s exact tests, and *p*-values for statistically significant differences (at *p* < 0.05) are shown in bold. The differences in prevalence between periods are provided as percentage points and the respective 95% confidence intervals. CLTI, chronic limb-threatening ischemia. ^A^ Antibiotics used in combination due to their synergistic effect.

**Table 4 antibiotics-13-00035-t004:** Clinically important antibiotic-resistant bacteria isolated from CLTI-associated leg ulcers before (1 January 2017–10 March 2020) and during (11 March 2020–31 December 2022) the COVID-19 pandemic.

Bacteria with Antibiotic Resistance	Overall	Pre-COVID-19	COVID-19
*n*	225	138	87
MRSA	1 (0.4%)	1 (0.7%)	–
MRSE	2 (0.9%)	–	2 (2.3%)
VRE	1 (0.4%)	1 (0.7%)	–
ESBL *Escherichia coli*	11 (4.9%)	9 (6.5%)	2 (2.3%)
ESBL *Klebsiella* spp.	7 (3.1%)	5 (3.6%)	2 (2.3%)
ESBL *Pseudomonas* spp.	1 (0.4%)	–	1 (1.1%)
ESBL *Proteus* spp.	8 (3.6%)	7 (5.1%)	1 (1.1%)
Carbapenem-resistant *Escherichia coli*	1 (0.4%)	–	1 (1.1%)
Carbapenem resistant *Klebsiella* spp.	1 (0.4%)	–	1 (1.1%)
Carbapenem resistant *Pseudomonas* spp.	2 (0.9%)	1 (0.7%)	1 (1.1%)
Carbapenem resistant *Proteus* spp.	3 (1.3%)	1 (0.7%)	2 (2.3%)
Carbapenem resistant *Acinetobacter* spp.	1 (0.4%)	2 (1.4%)	–
Multidrug resistant (MDR)	33 (14.7%)	26 (18.8%)	7 (8.0%) *

Data for the overall study or individual bacterial taxon or group represent the number and respective proportion out of the total number tested. ESBL, extended-spectrum β-lactamase-producing bacteria, including those resistant to third-generation cephalosporins, as defined by Thailand’s National Antimicrobial Resistance Surveillance Center; MDR, multidrug-resistant bacteria non-susceptible to at least one drug in three or more antibiotic categories, as defined by the Clinical Laboratory Standards Institute; MRSA, Methicillin-resistant *Staphylococcus aureus*; MRSE, Methicillin-resistant *Staphylococcus epidermidis*; VRE, Vancomycin-resistant *Enterococcus* spp.; * *p* = 0.026 for a difference in MDR prevalence between the two periods, derived from a Fisher’s exact test.

**Table 5 antibiotics-13-00035-t005:** Antibiotic resistance in gram-positive bacteria isolated from CLTI-associated leg ulcers before (1 January 2017–10 March 2020) and during (11 March 2020–31 December 2022) the COVID-19 pandemic.

Pathogens *n* (%)	Overall		Oxacillin		Ampicillin		Fusidic Acid		Clindamycin		Erythromycin		Trimethoprim/ Sulfamethoxazole ^A^		Penicillin		Aminoglycoside		Fluoroquinolones		Vancomycin	
	Pre	During	Pre	During	Pre	During	Pre	During	Pre	During	Pre	During	Pre	During	Pre	During	Pre	During	Pre	During	Pre	During
*Staphylococcus aureus* (*n* = 19; 40%)	**3/9** **(33%)**	**0/10 *** **(nil)**	1/9 (11%)	0/10 (nil)	ND		1/9 (11%)	0/10 (nil)	2/9 (22%)	0/10 (nil)	2/9 (22%)	0/10 (nil)	1/9 (11%)	0/10 (nil)	ND		0/9 (nil)	0/10 (nil)	0/9 (nil)	0/10 (nil)	0/9 (nil)	0/10 (nil)
*Enterococcus* spp. *(E. faecalis, E. faecium, E. avium)* (*n* = 17; 36%)	3/10 (30%)	3/7 (43%)	ND		1/10 (10%)	1/7 (14%)	ND		ND		ND		ND		1/10 (10%)	1/7 (14%)	2/10 (20%)	2/7 (29%)	0/10 (nil)	0/7 (nil)	1/10 (10%)	0/7 (0%)
*β*-hemolytic Streptococci (*n* = 7; 15%)	0/4 (nil)	0/3 (nil)	ND		0/4 (nil)	0/3 (nil)	ND		0/4 (nil)	0/3 (nil)	0/4 (nil)	0/3 (nil)	ND		0/4 (nil)	0/3 (nil)	0/4 (nil)	0/3 (nil)	0/4 (nil)	0/3 (0%)	0/4 (0%)	0/3 (0%)
*Staphylococcus epidermidis* (*n* = 2; 4%)	–	2/2 (100%)	–	2/2 (100%)	ND		ND		–	2/2 (100%)	–	2/2 (100%)	–	2/2 (100%)	ND		–	1/2 (50%)	–	1/2 (50%)	0/2 (0%)	–
*α*-hemolytic Streptococci (*n* = 1; 2%)	1/1 (100%)	–	ND		ND		ND		1/1 (100%)	–	1/1 (100%)	–	ND		ND		ND		ND		0/1 (nil)	–
*Staphylococcus haemolyticus* (*n* = 1; 2%)	–	0/1 (nil)	–	0/1 (nil)	ND		ND		–	0/1 (nil)	–	0/1 (nil)	–	0/1 (nil)	ND		–	0/1 (nil)	–	0/1 (0%)	–	0/1 (0%)
**Total** **(*n* = 47)**	7/24 (29%)	5/23 (22%)	1/9 (11%)	2/13 (15%)	1/14 (7%)	1/10 (10%)	1/9 (11%)	0/10 (0%)	3/14 (21%)	2/16 (13%)	3/14 (21%)	2/16 (13%)	1/9 (11%)	2/13 (15%)	1/14 (7%)	1/10 (10%)	2/23 (9%)	3/23 (13%)	0/23 (0%)	1/23 (4%)	1/26 (4%)	0/21 (0%)

CLTI, chronic limb-threatening ischemia; ND, no susceptibility testing performed. Data within cells are *n*/*N* (%), where *N* is the number of isolates tested against a given antibiotic (columns), *n* is the number with antibiotic resistance, and (%) is the corresponding proportion. ^A^ Antibiotics used in combination due to their synergistic effect. * *p* < 0.05 for the difference in antibiotic resistance prevalence between the two periods (shown in bold) derived from Fisher’s exact test. Note that comparisons were only examined for the overall counts for given bacterial taxa with isolates in both periods, the total counts for individual antibiotics, and the total overall.

**Table 6 antibiotics-13-00035-t006:** AR in gram-negative bacteria from CLTI-associated leg ulcers before (1 January 2017–10 March 2020) and during (11 March 2020–31 December 2022) the COVID-19 pandemic.

Bacterial Taxa	Overall		Carbapenems		Cephalosporins		Piperacillin/ Tazobactam ^A^		Fluoroquinolones		Aminoglycosides		Ampicillin		Amoxicillin/ Clavulanate ^A^		Trimethoprim/ Sulfamethoxazole ^A^	
	Pre	During	Pre	During	Pre	During	Pre	During	Pre	During	Pre	During	Pre	During	Pre	During	Pre	During
*Escherichia coli* (*n* = 27; 15%)	15/18 (83%)	8/9 (89%)	0/18 (nil)	1/9 (11%)	9/18 (50%)	2/9 (22%)	0/18 (nil)	1/9 (11%)	12/18 (67%)	7/9 (78%)	1/18 (6%)	3/9 (33%)	3/18 (17%)	0/9 (nil)	1/18 (6%)	1/9 (11%)	8/18 (44%)	4/9 (44%)
*Pseudomonas aeruginosa* (*n* = 26; 15%)	1/12 (8%)	1/14 (7%)	1/12 (8%)	1/14 (7%)	0/12 (nil)	1/14 (7%)	0/12 (nil)	0/14 (nil)	0/12 (nil)	1/14 (7%)	0/12 (nil)	1/14 (7%)	ND		ND		ND	
*Proteus mirabilis* (*n* = 17; 10%)	7/12 (58%)	2/5 (40%)	0/12 (nil)	0/5 (nil)	5/12 (42%)	1/5 (20%)	0/12 (nil)	0/5 (nil)	5/12 (42%)	2/5 (40%)	1/12 (8%)	1/5 (20%)	2/12 (17%)	0/5 (nil)	2/12 (17%)	0/5 (nil)	7/12 (58%)	2/5 (40%)
*Citrobacter freundii* (*n* = 16; 9%)	6/9 (67%)	2/7 (29%)	1/9 (11%)	0/7 (nil)	2/9 (22%)	0/7 (nil)	2/9 (22%)	1/7 (14%)	1/9 (11%)	1/7 (14%)	0/9 (nil)	1/7 (14%)	ND		0/7 (nil)	0/7 (nil)	2/9 (22%)	2/7 (28%)
*Klebsiella pneumoniae* (*n* = 15; 8%)	6/9 (67%)	4/6 (67%)	0/9 (nil)	1/6 (17%)	4/9 (44%)	2/6 (33%)	2/9 (22%)	0/6 (nil)	5/9 (55%)	4/6 (67%)	0/9 (nil)	0/6 (nil)	1/9 (11%)	0/6 (nil)	4/9 (44%)	0/6 (nil)	5/9 (55%)	4/6 (67%)
*Morganella morganii* (*n* = 13; 7%)	**10/10** **(100%)**	**1/3 *** **(33%)**	0/10 (nil)	0/3 (nil)	2/10 (20%)	0/3 (nil)	0/10 (nil)	0/3 (nil)	1/10 (10%)	1/3 (33%)	0/10 (nil)	0/3 (nil)	2/10 (20%)	0/3 (nil)	10/10 (100%)	0/3 (nil)	0/10 (nil)	1/3 (33%)
*Stenotrophomonas maltophila* (*e* = 9; 5%)	2/7 (29%)	1/2 (50%)	ND		0/7 (nil)	1/2 (50%)	ND		2/7 (29%)	0/2 (nil)	ND		ND		ND		0/7 (nil)	0/2 (nil)
*Serratia marcescens* (*n* = 8; 4%)	6/6 (100%)	0/2 (nil)	0/6 (nil)	0/2 (nil)	0/6 (nil)	0/2 (nil)	0/6 (nil)	0/2 (nil)	0/6 (nil)	0/2 (nil)	0/6 (nil)	0/2 (nil)	ND		6/6 (100%)	0/2 (nil)	0/6 (nil)	0/2 (nil)
*Proteus vulgaris* (*n* = 7; 4%)	1/3 (33%)	2/4 (50%)	0/3 (nil)	2/4 (50%)	1/3 (33%)	0/4 (nil)	0/3 (nil)	0/4 (nil)	1/3 (33%)	0/4 (nil)	1/3 (33%)	0/4 (nil)	1/3 (33%)	0/3 (nil)	0/3 (nil)	0/4 (nil)	1/3 (33%)	0/4 (nil)
*Enterobacter cloacae* (*n* = 6; 3%)	2/2 (100%)	1/4 (25%)	0/2 (nil)	0/4 (nil)	0/2 (nil)	0/4 (nil)	0/2 (nil)	0/4 (nil)	0/2 (nil)	1/4 (25%)	0/2 (nil)	0/4 (nil)	ND		2/2 (100%)	0/4 (nil)	0/2 (nil)	1/4 (25%)
*Klebsiella oxytoca* (*n* = 6; 3%)	2/5 (40%)	0/1 (nil)	0/5 (nil)	0/1 (nil)	1/5 (20%)	0/1 (nil)	0/5 (nil)	0/1 (nil)	2/5 (40%)	0/1 (nil)	0/5 (nil)	0/1 (nil)	ND		0/5 (nil)	0/1 (nil)	1/5 (20%)	0/1 (nil)
*Acinetobacter baumannii* (*n* = 4; 2%)	2/2 (100%)	0/2 (nil)	2/2 (100%)	0/2 (nil)	0/2 (nil)	0/2 (nil)	2/2 (100%)	0/2 (nil)	2/2 (100%)	0/2 (nil)	2/2 (100%)	0/2 (nil)	1/2 (50%)	0/2 (nil)	ND		0/2 (nil)	0/2 (nil)
Nonfermenting gram-negative bacilli (*n* = 4; 2%)	0/3 (nil)	1/1 (100%)	0/3 (nil)	0/1 (nil)	0/3 (nil)	0/1 (nil)	0/3 (nil)	0/1 (nil)	0/3 (nil)	1/1 (100%)	0/3 (nil)	1/1 (100%)	ND		ND		0/3 (nil)	0/1 (nil)
Others (*n* = 20; 11%) ^B^	9/16 (56%)	2/4 (50%)	2/16 (13%)	0/4 (nil)	1/15 (7%)	0/4 (nil)	0/16 (nil)	0/4 (nil)	4/16 (25%)	0/4 (nil)	2/16 (13%)	0/4 (nil)	1/5 (20%)	0/4 (nil)	6/15 (40%)	1/4 (25%)	0/15 (nil)	0/4 (nil)
**Total (*n* = 178)**	**69/114** **(61%)**	**25/64 **** **(39%)**	6/107 (6%)	5/62 (8%)	25/114 (22%)	7/64 (11%)	6/107 (6%)	2/62 (3%)	35/114 (31%)	18/64 (28%)	7/107 (6%)	7/62 (13%)	**11/59** **(19%)**	**0/33 **** **(nil)**	**37/89** **(42%)**	**2/45 ****** **(4%)**	24/101 (24%)	14/50 (28%)

CLTI, chronic limb-threatening ischemia; ND, no susceptibility testing performed. Data within cells are *n/N* (%), where *N* is the number of isolates tested against a given antibiotic (columns), *n* is the number with antibiotic resistance, and (%) is the corresponding proportion. ^A^ Antibiotics used in combination due to their synergistic effect. ^B^ This group included bacteria with fewer than 3 isolates, namely *Achromobacter xylosoxidans*, *Acinetobacter johnsonii*, *Aeromonas hydrophila*, *Citrobacter koseri*, *Delftia acidovorans*, *Haemophilus parainfluenzae*, *Proteus penneri*, *Providencia rettgeri*, *Providencia stuartii*, and *Pseudomonas putida*. * *p* < 0.05, ** *p* < 0.01, and **** *p* < 0.0001 for a difference in AR prevalence between the two periods (shown in bold) derived from Fisher’s exact tests. Note that comparisons were only examined for the overall counts for given bacterial taxa, the total counts for individual antibiotics, and the total overall.

**Table 7 antibiotics-13-00035-t007:** Risk factors for antibiotic resistance among the study population.

Parameter	OR (95% CI)	*p*	aOR (95% CI)	*p*
Study period (COVID-19 vs. prior)	0.21 (0.10, 0.46)	**<0.001**	0.21 (0.08, 0.51)	**0.001**
Age ≥ 60 years	0.98 (0.42, 2.30)	0.97		
Male	1.80 (0.87, 3.74)	0.11		
Diabetes mellitus	1.12 (0.53, 2.35)	0.77		
Renal replacement therapy	1.60 (0.69, 3.73)	0.28		
High fever (body temperature > 38 °C)	1.40 (0.56, 3.56)	0.47		
Heart rate (beats per minute)	1.09 (0.52, 2.30)	0.82		
Respiratory rate > 20 (breaths per min)	2.48 (0.66, 9.38)	0.18	6.17 (1.02, 37.3)	**0.047**
Leukocytosis (WBC > 12,000/mm^3^)	0.75 (0.36, 1.55)	0.44		
Recurrent ulcer on the same limb	1.05 (0.48, 2.27)	0.90		
Duration ulcer ≥ 3 months	1.51 (0.62, 3.64)	0.36		
Limb infection grade 3 ^A^	1.05 (0.50, 2.20)	0.89		
Wound grade 3 ^A^	1.01 (0.47, 2.16)	0.98		
Ischemic grade 3 ^A^	0.71 (0.34, 1.49)	0.37		
Presence of osteomyelitis	0.95 (0.44, 2.05)	0.91		
Any hospitalization within 6 months before admission	2.10 (1.01, 4.36)	**0.038**		
Referral from health services for CLTI	1.69 (0.76, 3.76)	0.20		
Polymicrobial infection	3.37 (1.59, 7.13)	**0.002**	5.58 (2.08, 15.0)	**0.001**
Gram-negative infection alone	2.57 (1.20, 5.52)	**0.015**	6.98 (2.38, 20.5)	**<0.001**
Mixed infection with gram-negative and gram-positive	1.41 (0.53, 3.74)	0.494		
Previous empirical antibiotic treatment	1.79 (0.79, 4.06)	0.16	11.9 (1.11, 128)	**0.041**

Data are the odds ratio (OR) or adjusted odds ratio (aOR) and their 95% confidence intervals derived from univariable or multivariable logistic regressions, respectively. Statistically significant associations (at *p* < 0.05) are shown in bold. CLTI, chronic limb-threatening ischemia; WBC, white blood cell count. ^A^ The SVS-WIfI classification grades CLTI based on perfusion (0–3), wound extent (0–3), and superadded infection (0–3), with 0 indicating absence and 3 indicating severe; the total score (summing these components) is associated with the risk of major amputations.

## Data Availability

The data supporting this study are available within the article. Raw data supporting this study’s findings are available from the corresponding author upon reasonable request, conditional on the necessary ethics approval of the statistical analysis plan.

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
