# Peer review of "A COVID-19 Silver Lining—Decline in Antibiotic Resistance in Ischemic Leg Ulcers during the Pandemic: A 6-Year Retrospective Study from a Regional Tertiary Hospital (2017–2022)"

_antibiotics, 2023, doi:10.3390/antibiotics13010035_

Round 1

Reviewer 1 Report

Comments and Suggestions for Authors

Authors examined the infection rates of antibiotic-resistant bacteria in ischemic leg ulcers among PAD patients admitted to a tertiary medical center before and during the COVID-19 pandemic.

Consider adding a sentence into abstract that emphasizes the practical implications of the findings and potential recommendations for clinical practice or public health interventions.

Please explain the choice of the hospital.

Were all samples collected before the initiation of antimicrobial treatment?

According to the figure 2 and table 3, each pathogen has a different resistance pattern and should not be compared all together, so it will be better first to explain identified pathogens and it resistance rate separately.

It is better to divide the results of sensitivity of different taxes shown in Table 6 according to the aim of the study: before and during Covid19.

Reviewer 2 Report

Comments and Suggestions for Authors

Point 1: Line 152 - "...specific criteria/conditions...." -> To what specific criteria or conditions does this refer?

Point 2: Line 168 - "...Methicillin-resistant Streptococcus epidermidis (MRSE)...." ->

Point 3: Line 171-172 - "...Pseudomonas spp., Proteus spp., and Acinetobacter spp...." -> Pseudomonas spp., and Acinetobacter spp do not belong to Enterobacteriaceae.

Point 4: In accordance with CLSI guidelines, certain bacteria such as Citrobacter freundii, Klebsiella pneumoniae, Morganella morganii, Serratia marcescens, Proteus vulgaris, Enterobacter cloacae, and Acinetobacter baumannii exhibit natural resistance to specific antibiotics. However, the author did not take this into consideration.

Point 5: Considering the inherent resistance observed, a reassessment of the antibiotic susceptibility statistics presented in the article is necessary.

Reviewer 3 Report

Comments and Suggestions for Authors

The authors present interesting results about antibiotic resistance in patients with ischemic leg ulcers during six years of retrospective data. In general, the manuscript is well-written and designed. However, it is reduced to only one health centre, and the same authors notice the weakness of the study. Despite that, the information shown could help other physicians or the scientific community in order to revise the enormous problem of bacteria resistance.

Only some minor mistakes should be fixed in the current version of the manuscript that is addressed here:

line 168 change "Streptococcus epidermidis"

Line 171 the authors mentioned Enterobacteriaceae, it could be better to mention this group of bacteria as Enterobacterales (current taxonomy for those bacteria). In addition, the authors mentioned Pseudomonas and Acinetobacter as members of "Enterobacteriaceae" this is wrong and must be the difference for those genera.

Line 339 write M. morganii in italic

Finally, in the discussion section, the authors could explain more about the empirical treatment vs the lab result of susceptibility of the isolates tested, due to is not discussed.

Round 2

Reviewer 1 Report

Comments and Suggestions for Authors

I do not have more comments.

Reviewer 2 Report

Comments and Suggestions for Authors

This revision has significantly improved the manuscript.I would recommend the manuscript for publication.